# Spexin-Mediated Dietary Adaptation in *Siniperca chuatsi*: Molecular Characterisation and Functional Insights into FABP2 Interaction

**DOI:** 10.3390/ani15202944

**Published:** 2025-10-10

**Authors:** Xiao Chen, Yunyun Yan, Junjian Dong, Hetong Zhang, Yuan Zhang, Fengying Gao, Xing Ye, Chengfei Sun

**Affiliations:** Pearl River Fisheries Institute, Chinese Academy of Fishery Sciences, Guangzhou 510380, China; cx1799069206@163.com (X.C.); yanyuny123@163.com (Y.Y.); dongjj@prfri.ac.cn (J.D.); zhanght@prfri.ac.cn (H.Z.); zjzhangyuan@prfri.ac.cn (Y.Z.); fengyinggao2011@163.com (F.G.); gzyexing@163.com (X.Y.)

**Keywords:** *Siniperca chuatsi*, *spexin*, glutathione S-transferase pull-down, fasting, large-body phenotype, FABP2

## Abstract

Growth performance represents a key metric in piscine genetic breeding. The growth-related gene spexin (*spx*) critically regulates appetite and metabolic homeostasis. This study characterized the open reading frame (ORF) of spx in mandarin fish (*Siniperca chuatsi*), an economically significant teleost, revealing its peak expression in the liver. We further elucidated the regulatory profile of *spx* during starvation and feeding adaptation. During a 7-day fasting regimen, *spx* expression in muscle, liver, and intestine exhibited an initial increase followed by a decline, whereas an inverse expression trajectory was noted in the brain and stomach. Larger individuals displayed significantly diminished *spx* expression in the liver and brain relative to their smaller counterparts, yet elevated expression was detected in the stomach. Using glutathione S-transferase (GST) pull-down assays, fatty acid-binding protein 2 was identified as a novel interaction partner of SPX, suggesting a potential mechanism whereby SPX modulates lipid metabolism via the peroxisome proliferator-activated receptor (PPAR) signaling pathway to mediate feeding adaptation. This study provides novel mechanistic insights into the role of SPX in nutritional adaptation in *S. chuatsi*, and establishes a foundational framework for subsequent genetic breeding initiatives.

## 1. Introduction

*Siniperca chuatsi* is an economically important freshwater species in East Asia, priced for its delicate flavour and boneless flesh, which drives high market demand. However, large-scale aquaculture of *S. chuatsi* is hindered by its obligate reliance on live prey, a consequence of its benthic, piscivorous feeding habits [1]. Although the development of formulated feeds has alleviated this dependency, considerable challenges persist, including low voluntary feed intake, substantial variation in growth performance, and hepatic metabolic dysfunction, which are key bottlenecks to production at a commercial scale. Starvation challenge trials are essential during the habituation of *S*. *chuatsi* to formulated feed. Studies have established the involvement of *spx* in fasting responses across fish species, but the specific regulatory mechanisms during fasting in mandarin fish need to be elucidated [2,3,4]. SPX is proposed to exert beneficial effects via multiple mechanisms: (1) inhibition of feeding behaviour through downregulation of orexigenic factors [5]; (2) improvement of hepatic lipid metabolism by modulating lipid-related enzyme and gene expression [6]; and (3) stimulation of lipolysis through enhanced phosphorylation of hormone-sensitive lipase (HSL) [7]. These mechanisms likely interact synergistically to maintain systemic metabolic homeostasis.

The *spx* gene was first identified in the human genome using bioinformatic analyses in 2007. Its cDNA was subsequently cloned in *Carassius auratus* and *Danio rerio* in 2013 [8] and later characterised in mammals, birds, and additional teleost species [9,10,11]. The genomic architecture of *spx* varies among teleosts. In both *C. auratus* and *D. rerio*, the gene comprises five exons and four introns. In *C. auratus*, alternative splicing generates three distinct transcripts (847 bp, 805 bp, and 574 bp), although only the 574 bp variant contains a complete open reading frame (ORF) encoding the mature SPX peptide. The remaining transcripts contain premature stop codons in the second intron, resulting in truncated, nonfunctional products [8,12]. In contrast, the *spx* genes of *S. chuatsi* and *Epinephelus coioides* exhibit a mammal-like structure, comprising six exons and five introns. This structural divergence is hypothesized to result from the insertion of a novel intron within the fifth exon [13,14].

Neuropeptide Q (spexin, SPX) has attracted attention for its distinct signalling features. The mature SPX peptide comprises a conserved 14-amino-acid core flanked by dibasic proteolytic cleavage sites [15]. Structural analyses in goldfish indicate two functional domains: Asn1–Pro4 and Gln5–Gln14 extending to the COOH terminus. The molecular surface exhibits high hydrophobicity. Lys11, the only charged residue, is essential for high-affinity receptor interaction [12]. SPX shares evolutionary ancestry with galanin (GAL) and kisspeptin (KISS), originating from a common ancestral peptide module in early vertebrates [16]. SPX activates GALR2 and GALR3 receptors in the central nervous system. Selective GALR2 activation engages the PI3K/AKT and L-type voltage-dependent calcium channel (VDCC) pathways, whereas binding to both GALR2 and GALR3 suppresses the cAMP/protein kinase A (PKA) signalling cascade [17,18]. Disruption of GALR2/3 signalling in hypothalamic nuclei, such as the arcuate nucleus (ARC), ventromedial hypothalamus (VMH), and dorsomedial hypothalamus (DMH), induces a positive energy balance, leading to glucose intolerance, lipid accumulation, and weight gain [19]. However, the structure and function of the *spx* gene in *S. chuatsi* remain uncharacterized. Furthermore, although the role of spexin in feeding is well-established, its direct influence on growth variation and lipid metabolism in teleosts is poorly understood, particularly on a mechanistic basis. To investigate the potential role of the *spx* gene (*Scspx*) in nutrient metabolism regulation in mandarin fish (*Siniperca chuatsi*), we cloned *Scspx* from feed-adapted fish, profiled its tissue expression patterns, and examined its expression in response to starvation and in populations with divergent feed-adaptation phenotypes. Furthermore, we screened for potential interacting proteins to elucidate the regulatory function of *Scspx* in feeding and metabolic pathways during the feed adaptation process.

## 2. Materials and Methods

### 2.1. Animals

Mandarin fish (*S. chuatsi*) fingerlings (53.12 ± 10.35 g; SL, ~12 cm; 5-month-old), designated for cloning, tissue expression profiling, and starvation experiments, were cultured for 120 d in aquaculture ponds (78 m × 36 m × 2.3 m) at Heyang Aquatic Co., Ltd. (Foshan, China). During this period, they were fed a formulated floating feed purchased from CP Group (diameter: 0.55 cm, crude protein ≥ 60%; total phosphorus ≥ 1.8%; crude fibre ≤ 7.0%; calcium ≤ 0.6–2.0%; crude fat ≥ 5.0%; lysine ≥ 1.8%; coarse grey powder ≤ 13.0%; moisture content ≤ 12.0%). The daily feed ration for each experimental group was adjusted based on the individual size of the mandarin fish: a feeding rate of 4.2% of body weight per day was applied during the early grow-out phase (50–200 g), and a rate of 2.8% of body weight was used during the late grow-out phase (>200 g).

A net cage (1.8 m × 1.8 m × 1.4 m; 0.3 cm mesh size) was installed within the pond, and 25 individuals were randomly selected from the main stock and transferred into the cage. Healthy mandarin fish were selected from net-pen cultured populations. Following dissection and sex identification, the subjects were randomly assigned to groups with a 1:1 male-to-female ratio, thereby minimizing any confounding effects attributable to sex. To minimize the potential influence of circadian rhythms on spexin gene expression, the sampling time for all experimental groups (including the various fasting groups and the control group) was strictly restricted to between 9:00 and 10:00 a.m. each day. Experimental groups underwent starvation conditioning with sampling at 3, 5, and 7 days post-starvation, with five fish randomly collected from the net cage at each time point. A total of five mandarin fish (*S. chuatsi*) maintained under routine feeding conditions were selected from the pond. After a 30 min feed withdrawal, surplus feed was removed upon cessation of feeding activity to establish a fasting condition for the net-caged cohort. Prior to sampling, all mandarin fish in net cages were confirmed to have ventral concavity, and the absence of intestinal contents was verified to strictly ensure that they were in a fasting state. The caged individuals underwent a 3 d acclimatization period prior to the initiation of complete feed deprivation to prevent stress-induced mortality, whereas the pond population continued daily feeding. No mortality was observed during the culture period.

To examine phenotypic variation in feed adaptation, ten feed-trained overwintering *S. chuatsi* individuals were collected from a commercial aquaculture facility in Longjiang, Shunde (Guangzhou, China). Sexually mature fish were classified into two groups based on body size: a large-body phenotype group (798.0 ± 23.10 g; n = 5; 14-month-old) and a small-body phenotype group (336.67 ± 14.70 g; n = 5; 14-month-old). Each phenotype cohort (large-body and small-body) included representative males and females demonstrating differential adaptation to feeding regimes.

All experiments were carried out in accordance with the “Guidelines for the Protection and Use of Laboratory Animals in China”. All experimental procedures and sample collection were approved by the Animal Experimentation Ethics Committee of the Pearl River Fisheries Research, Chinese Academy of Fishery Sciences.

### 2.2. Cloning of the Scspx ORF Sequence

The ORF of the *spx* gene was identified from the *S. chuatsi* genome. Two specific primer pairs (*spx*-F1/R1 and *spx*-F2/R2; Table 1) were designed using the PrimerQuest Tool (accessed on 25 September 2024) and synthesized by Sangon Biotech (Shanghai, China). Total RNA was extracted from liver tissue using TRIzol^®^ Reagent (Ambion, Austin, TX, USA), and first-strand cDNA was synthesized using the ToloScript RT EasyMix for qPCR Kit (Novoprotein, Shanghai, China). Gene amplification was performed using a nested PCR approach. PCR products were separated on 1.5% agarose gels, and bands of the expected size were excised and purified using the Wizard^®^ SV Gel and PCR Clean-Up System (Promega, Madison, WI, USA).

Purified fragments were ligated into the pBM23 vector via one-step TOPO-TA cloning at 25 °C for 4 h. Positive transformants were identified using colony PCR. Plasmids containing inserts of the expected size were confirmed using Sanger sequencing (IGE Biotechnology Ltd., Guangzhou, China).

### 2.3. SPX Protein Sequence Analysis in S. Chuatsi

The *Scspx* ORF sequence was translated into its corresponding amino acid sequence using DNAMAN software 8.0 version (Lynnon Biosoft). Homology searches were conducted via BLAST (-2.14.0) against the National Center for Biotechnology Information (NCBI) non-redundant protein database. Prior to analysis, homogeneity of variance analysis was performed, and outliers were removed to minimize experimental error. Multiple sequence alignments of SPX peptides from 12 teleost species were performed using ClustalW, with reference sequences obtained from GenBank.

Physicochemical properties, conserved domains, and predicted tertiary structures of SPX were analysed using ExPASy ProtParam (accessed on 5 October 2024), SMART(accessed on 5 October 2024), and SPOMA, respectively. A phylogenetic tree was constructed using the neighbour-joining method in MEGA 11. All quantitative data are presented as mean ± standard error. Statistical analyses were performed using SPSS 21.0 (SPSS, Armonk, NY, USA), and differences were considered significant at *p* < 0.05.

### 2.4. qRT-PCR Detection of spx Tissue Expression Profiles

The transcript abundance of the *spx* gene in various tissues of *S. chuatsi* was assessed using quantitative real-time PCR (qRT-PCR). *β-Actin* was used as the internal reference gene. Primers targeting *spx* (q*spx*-F/R) were designed and synthesized by Sangon Biotech (Shanghai, China; Table 1). Five healthy *S. chuatsi* individuals (238.2 ± 51.39 g; 8-month-old) were euthanised following approved ethical protocols. Approximately 20 mg of tissue from the liver, muscle, brain, stomach, intestine, and gill was aseptically collected using autoclaved instruments. Total RNA extraction and cDNA synthesis were performed as previously described. The qRT-PCR was conducted using the LightCycler^®^ 96 System (Roche, Shanghai, China). Relative gene expression levels were quantified using the 2^−ΔΔCt^ method.

### 2.5. spx Gene Expression During Fasting and Feed Adaptation

To investigate temporal *spx* expression during fasting and dietary adaptation, *S. chuatsi* individuals were euthanised from both net-cage (fasting 3, 5, and 7 d, n = 5 per time point) and pond-fed (control) groups. Tissue samples (~20 mg each) from the telencephalon, liver, intestine, skeletal muscle, and gastric epithelium were collected under aseptic conditions, immediately snap-frozen in liquid nitrogen, and stored at −80 °C. The qRT-PCR was used to analyse *spx* expression at 3, 5, and 7 d post-treatment. Total RNA was isolated, reverse transcribed, and subjected to qRT-PCR as described above. Data were statistically analysed using one-way analysis of variance in GraphPad Prism 10 version (GraphPad Software, San Diego, CA, USA), followed by Fisher’s least significant difference test for multiple comparisons. An independent-sample t-test was used to compare the differences in *spx* expression between the large- and small-body size groups of *S. chuatsi*, differences were considered significant at *p* < 0.05. The relative expression of *spx* in liver and brain tissues was compared between large-body and small-body phenotype groups to assess differential expression associated with feed adaptation.

### 2.6. GST Pull-Down Assay

The ORF of *spx* was amplified from liver-derived cDNA of feed-trained *S. chuatsi* and cloned into the pGEX-6P-1 vector (digested with BamHI and XhoI). The recombinant plasmid (pGEX-6P-SPX) was transformed into *Escherichia coli* BL21 (DE3) cells. Positive clones were induced using isopropyl β-D-1-thiogalactopyranoside (IPTG)and cultured for 8 h. Bacterial pellets were resuspended in STET buffer containing lysozyme, lysed by sonication, and centrifuged. The resulting supernatant was purified using a glutathione S-transferase (GST) affinity column. Protein purity was verified using sodium dodecyl sulphate–polyacrylamide gel electrophoresis (SDS-PAGE). Total protein was extracted from the liver of feed-ingesting *S. chuatsi* and examined using Western blot assays with a 12% SDS-PAGE gel. Proteins were transferred onto polyvinylidene difluoride membranes, blocked overnight at 4 °C, and sequentially incubated with sheep anti-GST (1:10,000), followed by anti-rabbit IgG secondary antibody. Signals were visualised using the Clarity Max™ Western ECL substrate (Bio-Rad, Hercules, CA, USA).

A total of 500 μg of GST–SPX fusion protein (experimental group) or GST alone (control group) was incubated with magnetic beads for 2 h at room temperature. Liver total protein extract was added to each group and adjusted to 1 mL with immunoprecipitation (IP) dilution buffer, followed by incubation at 4 °C for 1 min. Subsequently, 100 μL of elution buffer was added to each group and incubated in a 95 °C water bath for 5 min. After centrifugation at 12,000× *g* for 5 min, 20 μL of 6× loading buffer was added to the supernatants. For the input group, 100 μL of liver total protein was mixed with 20 μL of 6× loading buffer and denatured in the same way. Western blotting was used to assess protein profiles in the marker, control, experimental, and input groups. Following SDS-PAGE and silver staining, protein bands were excised and subjected to in-gel tryptic digestion. Proteins were reduced, alkylated, digested overnight with trypsin, and desalted using an SDB column. Peptides were analysed using an UltiMate 3000 RSLCnano system coupled with a Q Exactive HF mass spectrometer (Thermo Fisher Scientific, Waltham, MA, USA). Raw MS data were searched against the UniProt database using MaxQuant (v2.2.0.0). Differentially interacting proteins were identified and annotated using Gene Ontology (GO) analysis.

## 3. Results

### 3.1. Sequence Analysis of spx in S. chuatsi

Specific primers were designed based on the *S. chuatsi* genome to amplify a 312 bp ORF encoding a 103-amino acid protein with an ATG start codon and a TGA stop codon. SignalP 4.1 analysis predicted the absence of a signal peptide in the SPX protein; however, a 14-amino acid mature peptide (NWTPQAMLYLKGTQ) was identified. Physicochemical characterisation using the ExPASy ProtParam tool indicated that the SPX protein had a molecular formula of C_536_H_817_N_151_O_162_S_2_, a molecular weight of 12.03 kDa, and a theoretical isoelectric point (pI) of 5.81.

Phylogenetic analysis was conducted using MEGA 11.0, incorporating SPX amino acid sequences from *S. chuatsi* and 11 other teleost species obtained from the NCBI database. The resulting neighbour-joining phylogenetic tree resolved two major clades. *S. chuatsi* clustered most closely with *Oncorhynchus mykiss* and *Cynoglossus semilaevis*, and more distantly from *D. rerio* and *Acipenser baerii* (Figure 1).

### 3.2. Tissue Distribution of Scspx

To investigate the tissue-specific expression of *Scspx*, qRT-PCR was conducted on six tissues from *S. chuatsi* reared on formulated feeds. Normalised expression data indicated ubiquitous *spx* expression, with the highest levels detected in liver tissue and the lowest in muscle. Hepatic *spx* expression was 17.36-fold higher than that in muscle. However, *spx* mRNA levels in the liver and other tissues, excluding muscle, were similar (*p* > 0.05) (Figure 2).

### 3.3. Regulation of spx Expression During Fasting and Feed Adaptation

To investigate *spx* transcriptional dynamics during fasting and in individuals with divergent growth phenotypes, qRT-PCR was used to quantify *spx* expression in liver, brain, and stomach tissues. During fasting (0, 3, 5, and 7 d), the liver exhibited the highest *spx* expression, whereas the stomach had the lowest. A conserved temporal expression pattern was observed across tissues, characterised by an initial increase followed by a decline. In the liver, *spx* expression peaked at day 3 and was lowest on day 5. Intestinal and gastric *spx* levels on day 7 exceeded those on day 3 (Figure 3). In the feed adaptation studies, individuals with the largest body size had substantially lower *spx* expression in the liver and brain compared to the smallest group, whereas expression in the stomach was elevated (*p* < 0.05) (Figure 4).

### 3.4. Identification of Expression Vectors and Purification of Fusion Proteins

To identify SPX-interacting proteins in *S. chuatsi*, GST pull-down assays were conducted using Mag-Beads conjugated with GST-tagged fusion proteins. Liver tissue lysates from individuals fed a standard formulated diet served as the protein source. The ORF of *Scspx* (324 bp) was inserted into the pGEX-6P-1 vector (4984 bp), yielding the recombinant plasmid, pGEX-6P-SPX (5308 bp). DNA sequencing confirmed the correct insertion of the *Scspx* sequence, verifying the successful construction of the expression vector.

Following transformation into *Escherichia coli* BL21, expression of the GST–SPX fusion protein was induced by isopropyl β-D-1-thiogalactopyranoside (IPTG). SDS-PAGE analysis indicated a distinct band at ~38 kDa, corresponding to the expected size of the GST–SPX fusion protein (Figure 5). The fusion protein was purified using GST affinity chromatography. Elution fractions exhibited markedly improved purity on SDS-PAGE, confirming successful enrichment of the recombinant GST-SPX protein.

### 3.5. GST Pull-Down Identification and Characterisation of SPX-Interacting Proteins

Total liver proteins were extracted from *S. chuatsi* specimens adapted to artificial feed. Western blot analysis confirmed uniform protein distribution and suitability for subsequent pull-down assays. The GST–SPX fusion protein and a control GST protein were incubated with Mag-Beads, alongside liver protein extracts from the same dietary group. The experimental group consisted of GST–SPX + Mag-Beads + total protein; the control group consisted of GST + Mag-Beads + total protein. Following incubation and washing, bound proteins were eluted and analysed using SDS-PAGE, silver staining, and liquid chromatography–tandem mass spectrometry (LC–MS/MS). Differential bands, those not attributable to nonspecific GST-tag interactions, were further verified using Western blot with anti-GST antibodies. Under stringent identification criteria (≥2 unique peptides per protein), a total of 63 high-confidence SPX-interacting proteins were detected in the experimental group, comprising 17 annotated proteins and 46 uncharacterised proteins. By comparison, 74 proteins were identified in the control group, with 58 proteins common to both. Detailed information on SPX-specific interactors is summarised in Table 2.

To functionally annotate the putative SPX-binding proteins, GO enrichment analysis was performed across three primary categories: molecular function (MF), cellular component (CC), and biological process (BP). A total of 21 significantly enriched Level 2 GO terms were identified. Interacting proteins were predominantly associated with ribosomal structure and binding activities, including RNA and adenyl nucleotide binding. These proteins were localised in the cytoplasm, cytosol, and ribonucleoprotein complexes, and were implicated in critical processes, such as translation, protein folding, and cellular homeostasis (Figure 6).

## 4. Discussion

The SPX prepropeptide conforms to the structural characteristics of a canonical secretory protein, comprising a conserved 14-amino-acid mature peptide with C-terminal amidation and several non-conserved signal peptides. A paralogous gene, designated *spx2*, has been identified in non-mammalian vertebrates [16], whereas the original *spx* was subsequently renamed *spx1*. To date, *spx2* has not been identified in mammalian genomes. Zhao et al. (2022) cloned the full-length cDNA of *spx2* in *D. rerio*, indicating a 288 bp ORF encoding a 95-amino-acid precursor protein [20]. The mature SPX2 peptide, characterised in *Oryzias latipes*, *Latimeria chalumnae*, and *D. rerio*, shares a conserved sequence: NWGPQSMLYLKGRYGR [16]. Investigation of the *spx* gene in *S*. *chuatsi* led to the cloning of its ORF. Comparative analysis determined that the deduced protein sequence is highly conserved across teleost fish, with maximum sequence identity observed in *O*. *niloticus.* Furthermore, based on conserved domain analysis, the gene was definitively assigned to the *spx1* subtype.

The *spx1* is broadly expressed in the central nervous system, peripheral tissues, and reproductive organs across species, such as humans, rodents, and goldfish, and is found in the cerebral cortex, liver, and ovary, *inter alia* [9]. This widespread and evolutionarily conserved expression pattern suggests multifunctional regulatory roles for *spx1*. Expression levels of the *spx* gene vary markedly across species and tissues. In *A. baerii*, *spx* mRNA is predominantly detected in the hypothalamus, intestine, and liver [10], whereas in *E. coioides*, high expression is observed in the brain, liver, and ovary, with minimal expression in the intestine [14]. In *D. rerio*, in situ hybridisation indicated that *spx* was expressed in the mesencephalon and metencephalon, whereas *spx2* was localised to the preoptic hypothalamus [21], indicating that the two paralogs may serve distinct physiological functions. In *S. chuatsi*, qRT-PCR analysis showed abundant *spx* expression in the liver, intestine, and telencephalon under both basal and fasting conditions. This tissue distribution suggests a role in regulating digestion, metabolism, and neural activity, aligning with observations in *A. baerii*, *E. coioides*, and rodents [10,11,14]. Interspecific variability in tissue-specific expression is evident [10]. For instance, in *Scatophagus argus*, *spx* is most highly expressed in the ovary, with minimal expression in the liver and brain [4]. Similarly, Kim et al. (2019) localised *spx* expression in *D. rerio* to the mesencephalon and metencephalon [21]. The observed variations in *spx* expression across species and tissues may be attributed to ecological adaptations associated with energy metabolism. As typical carnivorous fish, *S. chuatsi* and *A. baerii* exhibit metabolic traits adapted to high-protein and high-lipid diets, resulting in elevated energy metabolic demands on organs such as the liver and intestine. The high expression of *spx* in these tissues aligns with its putative roles in appetite suppression and lipid metabolism regulation, suggesting an evolutionary adaptation to a carnivorous ecological niche. In contrast, the omnivorous *S. argus* has a broader dietary spectrum, which may have driven the evolution of distinct energy utilization strategies. Such ecological divergence likely underlies the differential *spx* expression profiles: in *S. argus*, *spx* expression is diminished in conventional metabolic tissues but enhanced in regions associated with other physiological processes. These divergent expression profiles suggest species-specific adaptation of *spx* function to distinct physiological roles.

In mammals, *spx* has been implicated in regulating intestinal motility, glucose and lipid metabolism, and hepatic function [22]. Functional studies have shown that SPX modulates glycemia by promoting insulin release, influences gastric contractions, and regulates adrenal cortex cell proliferation [2,9,18,23]. However, several SPX functions characterised in fish, such as suppression of luteinizing hormone (LH), appetite inhibition, and energy balance regulation, are not conserved in mammals. Fasting experiments in *E. coioides*, *S. argus*, and *Cynoglossus semilaevis* have consistently shown substantial upregulation of hypothalamic *spx* expression in response to nutrient deprivation [4,12,24].

The present study demonstrated that *spx* expression in *S. chuatsi* liver was substantially elevated during the early phase of fasting. We propose that this upregulation contributes to appetite suppression and reduced energy intake, thereby promoting metabolic homeostasis under starvation conditions. Supporting this, *spx1* knockout models in zebrafish have shown that SPX mediates appetite inhibition via suppression of *AgRP1* expression [24]. Moreover, SPX–insulin interactions within islet cells establish a paracrine feedback loop modulating insulin secretion [9]. The findings of the current study suggest that SPX plays a conserved role in maintaining energy balance during fasting. In *S. chuatsi*, the observed downregulation of neural *spx* expression during prolonged starvation (5 d) may reflect an adaptive shift in response to sustained energy deficiency.

Observations during the domestication of *S*. *chuatsi* to formulated feed indicated marked phenotypic variation among individuals, despite overall successful domestication. Although some individuals exhibited robust growth performance and good feed adaptation, others showed signs of growth retardation and poor adaptation to the feed. Although multiple studies have established that the *spx* gene regulates feeding behaviour and satiety in teleosts [2,10,12], its role in the dietary adaptation of *S. chuatsi* remains incompletely understood. The present study provides the first experimental evidence supporting *spx* involvement in the dietary adaptation of *S. chuatsi*. Comparative expression analysis indicated substantially lower *spx* levels in the liver and brain of large-size individuals relative to small-size counterparts, whereas *spx* expression in the stomach was markedly upregulated in the larger group.

Using *C. auratus* as a model, Wong et al. (2013) demonstrated that *spx* suppressed food intake via insulin-mediated signalling in hepatic and cerebral tissues [12]. In vitro experiments indicated a dual regulatory mechanism of insulin on *spx* expression: insulin acts locally as an autocrine/paracrine signal to stimulate *spx* expression, and systemically as an endocrine factor to induce cerebral *spx* transcription [12]. Similarly, in *E. coioides*, peripheral administration of SPX-14 substantially upregulated *pro-opiomelanocortin (pomc)* expression while downregulating *growth hormone (gh)* and *orexin* mRNA levels in pituitary cells [14]. In *D. rerio*, *spx1* knockout models exhibited markedly increased food intake compared with wild-type controls. Intracranial injection of SPX1 suppressed *agouti-related peptide 1 (AgRP1)* expression, implicating SPX1 as a satiety signal that exerts appetite-suppressive effects via downregulation of *AgRP1* [5]. In mammals, Mirabeau et al. (2007) localised *spx* expression to the oesophageal and gastric submucosa, where the propeptide was shown to induce gastric smooth muscle contraction [25]. In contrast, injection of SPX in *A. baerii* paradoxically led to a reduction in endogenous *spx* expression, while simultaneously increasing mRNA levels of *nucleobindin 2 (nucb2)* and *peptide YY (PYY)*, consistent with an anorexigenic regulatory role of SPX in sturgeon [9]. Collectively, these findings suggest that downregulation of *spx* in the liver and brain of large-sized *S. chuatsi* may facilitate increased feed intake by modulating hormonal profiles and repressing anorexigenic gene expression. Conversely, elevated *spx* expression in the stomach may enhance digestive efficiency by promoting smooth muscle contractility, thereby indirectly contributing to growth performance.

Fatty acid-binding proteins (FABPs) represent a conserved multigene family of intracellular lipid-binding proteins (iLBPs), derived via duplication and diversification of an ancestral *ilbp* gene [26]. To date, 12 distinct FABP isoforms have been identified, which reversibly bind hydrophobic ligands and facilitate their transport between cellular compartments to support diverse physiological processes [27,28]. Kaitetzidou et al. established a fasting-refeeding experimental model in gilthead sea bream and European sea bass, systematically revealing a dynamic regulatory relationship between FABP2 gene expression and nutritional status in fish [29]. The study demonstrated that fasting significantly down-regulated FABP2 mRNA expression in the intestinal tissues of both species, while refeeding promptly restored expression to baseline levels. This dynamic expression pattern indicates that FABP2 is directly regulated by the nutritional state of the organism. Notably, this regulatory mechanism appears highly conserved across teleost species. Similarly, Xia et al. observed in Asian sea bass that short-term fasting (3d, 6d, 12d) markedly suppressed the expression of both *fabp2a* and *fabp2b* genes in the intestine [30]. Furthermore, studies in zebrafish confirmed that increasing dietary lipid content significantly up-regulated intestinal *fabp2* transcript levels. The results suggest that the regulation of *fabp2* gene transcription by fatty acids is mediated by the interaction of the peroxisome proliferator-activated receptor (PPAR) with a peroxisome proliferator response element (PPRE) in its promoter region [31]. These cross-species findings collectively suggest that FABP2 plays a critical role in nutrient metabolism in teleosts, likely participating in the regulation of lipid metabolic pathways through similar molecular mechanisms across different fish species, thereby supporting the view that this protein serves a core function in lipid absorption and transport. Based on the observed upregulation of *spx* expression in the intestine of *S. chuatsi* subjected to short-term starvation after feeding, this study proposes a regulatory hypothesis wherein *spx* may indirectly facilitate *fabp2* gene transcription via the upregulation of PPAR-α expression. This signalling cascade potentially represents a key mechanistic axis through which *spx* exerts its pivotal role in modulating lipid metabolism in fish.

In the current study, GST pull-down assays confirmed a specific interaction between SPX and FABP2. Further analysis indicated that FABP2 was enriched within the peroxisome proliferator-activated receptor (PPAR) signalling pathway, which governs lipid oxidation, adipocyte differentiation, hepatic lipid metabolism, cellular proliferation, and glucose uptake [32]. Zheng et al. demonstrated that SPX positively upregulated hepatic *AdipoQ* expression via activation of the PLC/PKC-Ca^2+^/CaMKII-MEK/ERK signalling pathway, thereby enhancing postprandial satiety. Conversely, adiponectin (*AdipoQ*) negatively suppressed SPX expression through the AMPK/PPAR-PI3K/Akt-p38 MAPK pathway, establishing a self-terminating, localized feedback loop within the liver [33]. As the most abundant circulating adipokine, adiponectin (*AdipoQ*) critically governs lipid/glucose metabolism and may directly influence feeding regulation [34]. In Siberian sturgeon (*Acipenser baerii*), intraperitoneal administration of recombinant globular adiponectin protein (*SsgAd*) acutely inhibited feeding. This effect was mediated by upregulation of the valvular intestinal anorexigenic peptide, *PYY*; modulation of hypothalamic appetite-regulating genes (*POMC*, *NPY*, *AGRP*, and *CART*); and activation of the hypothalamic AMPK/mTOR signalling pathway. Specifically, SsgAd administration promoted the expression of *AdipoR1*, *AMPKα2*, *Akt*, and *mTOR*, while suppressing the expression of *AMPKβ1*, *AMPKβ2*, and *AMPKγ2* [35]. The PPAR signalling cascade is regulated downstream of the AMP-activated protein kinase (AMPK) pathway. Phosphorylated AMPK stimulates PPARα, which upregulates genes involved in fatty acid transport and mitochondrial *β*-oxidation, whereas PPAR-γ activation enhances insulin sensitivity and facilitates glucose uptake [36]. Acting as a cellular energy sensor, AMPK responds to increased AMP/ATP ratios by activating catabolic pathways, such as fatty acid oxidation and autophagy, and suppressing anabolic processes, such as fatty acid and cholesterol biosynthesis [37].

Thus, we hypothesized that the bidirectional regulation between SPX and AdipoQ activated AMPK, triggering a cascade of AMPK-signalling-regulated reactions that further modulated FABP2 in the PPAR signalling pathway. This SPX–AMPK–PPAR–FABP2 axis may represent a key regulatory route through which SPX influences lipid metabolism and energy homeostasis in *S. chuatsi*. While the current study provides compelling evidence for the association between *Scspx* expression and growth characteristics in *S. chuatsi*, it is important to acknowledge its limitations. The functional inferences regarding *Scspx* are primarily derived from correlative observations in expression profiling. To advance from correlation to causation, direct functional validation is essential. Future investigations utilizing gene knockout/knockdown or overexpression approaches, both in vitro and in vivo, will be crucial to unequivocally establish the physiological role of *Scspx* in regulating lipid metabolism and its interaction with FABP2. Such studies will ultimately determine whether the observed expression changes represent a cause or a consequence of the metabolic alterations.

## 5. Conclusions

Previous genome-wide association analyses identified the *spx* gene as being closely associated with feed adaptation in *S. chuatsi*. In the current study, the ORF of the *spx* gene was successfully cloned and characterised in feed-trained *S. chuatsi*. Transcriptional profiling indicated that *spx* expression was modulated by fasting and was associated with the degree of feed adaptation. Furthermore, in vitro GST pull-down assays demonstrated a direct interaction between SPX and FABP2 proteins. These findings provide novel insights into the functional role of ScSPX in the PPAR signalling pathway and its potential involvement in the feed domestication process of *S. chuatsi*.

## Figures and Tables

**Figure 1 animals-15-02944-f001:**
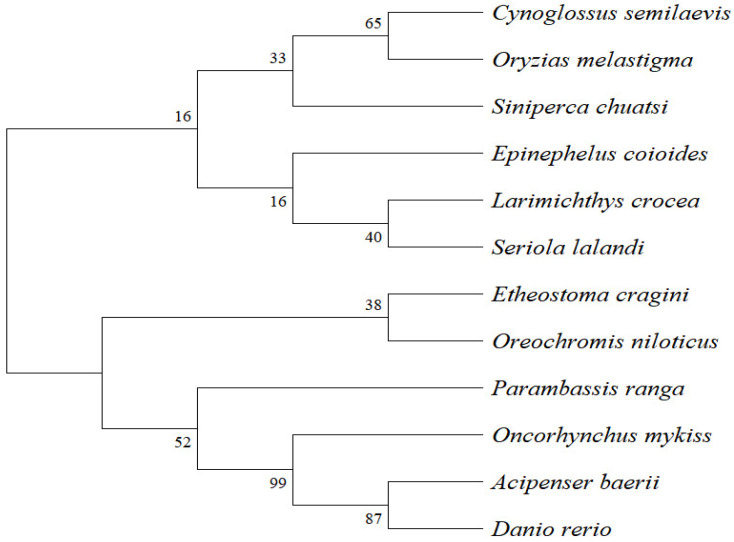
Phylogenetic tree of SPX protein sequences constructed using MEGA 11.0 (neighbour-joining method).

**Figure 2 animals-15-02944-f002:**
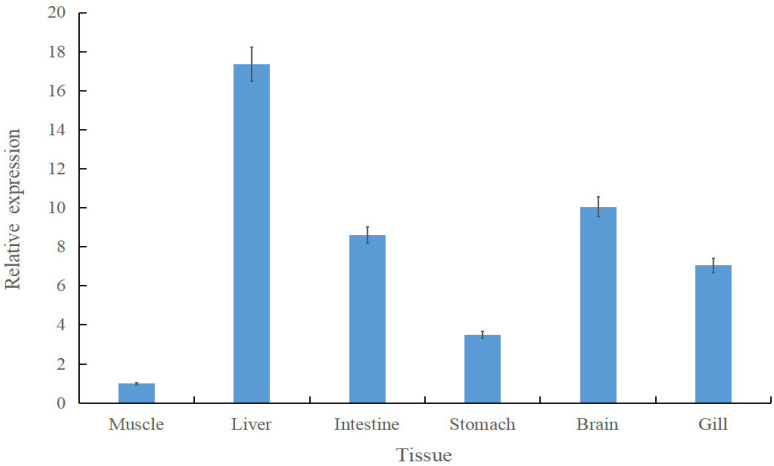
Tissue-specific expression of *Scspx* in *Siniperca chuatsi*. Data are presented as mean ± standard deviation.

**Figure 3 animals-15-02944-f003:**
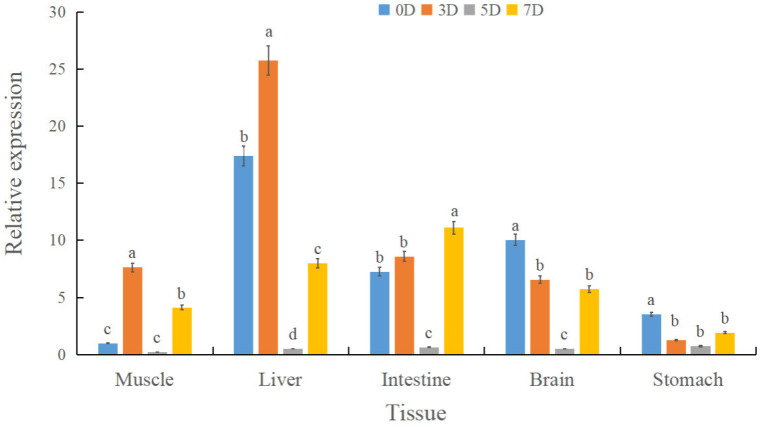
Relative *spx* expression in liver, brain, stomach, and intestine of *Siniperca chuatsi* under graded starvation (0, 3, 5, and 7 d). Different letters (a, b, c, d) above the bars indicate statistically significant differences among groups as determined by a one-way analysis of variance (ANOVA). The assignment of letters was performed such that groups sharing a common letter are not significantly different from one another at the *p* > 0.05 level. Conversely, groups labeled with different letters exhibit a statistically significant difference (*p* < 0.05). The comparisons were conducted across all groups simultaneously.

**Figure 4 animals-15-02944-f004:**
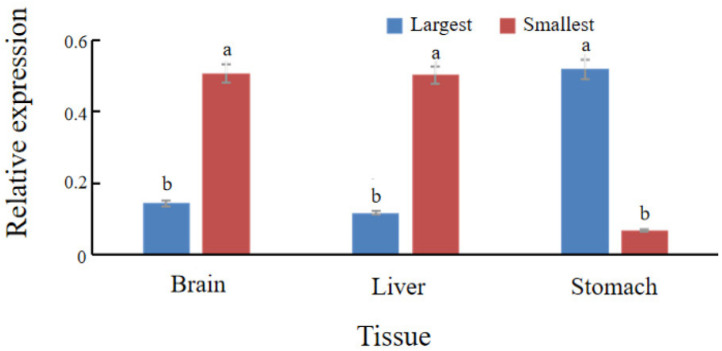
Comparative *spx* expression in liver, brain, and stomach tissues between the largest and smallest size groups of *S. chuatsi* reared on formulated feeds. Mean values assigned different letters (a, b) differ significantly (*p* < 0.05). The absence of a common letter indicates a statistically significant difference in the pairwise comparison.

**Figure 5 animals-15-02944-f005:**
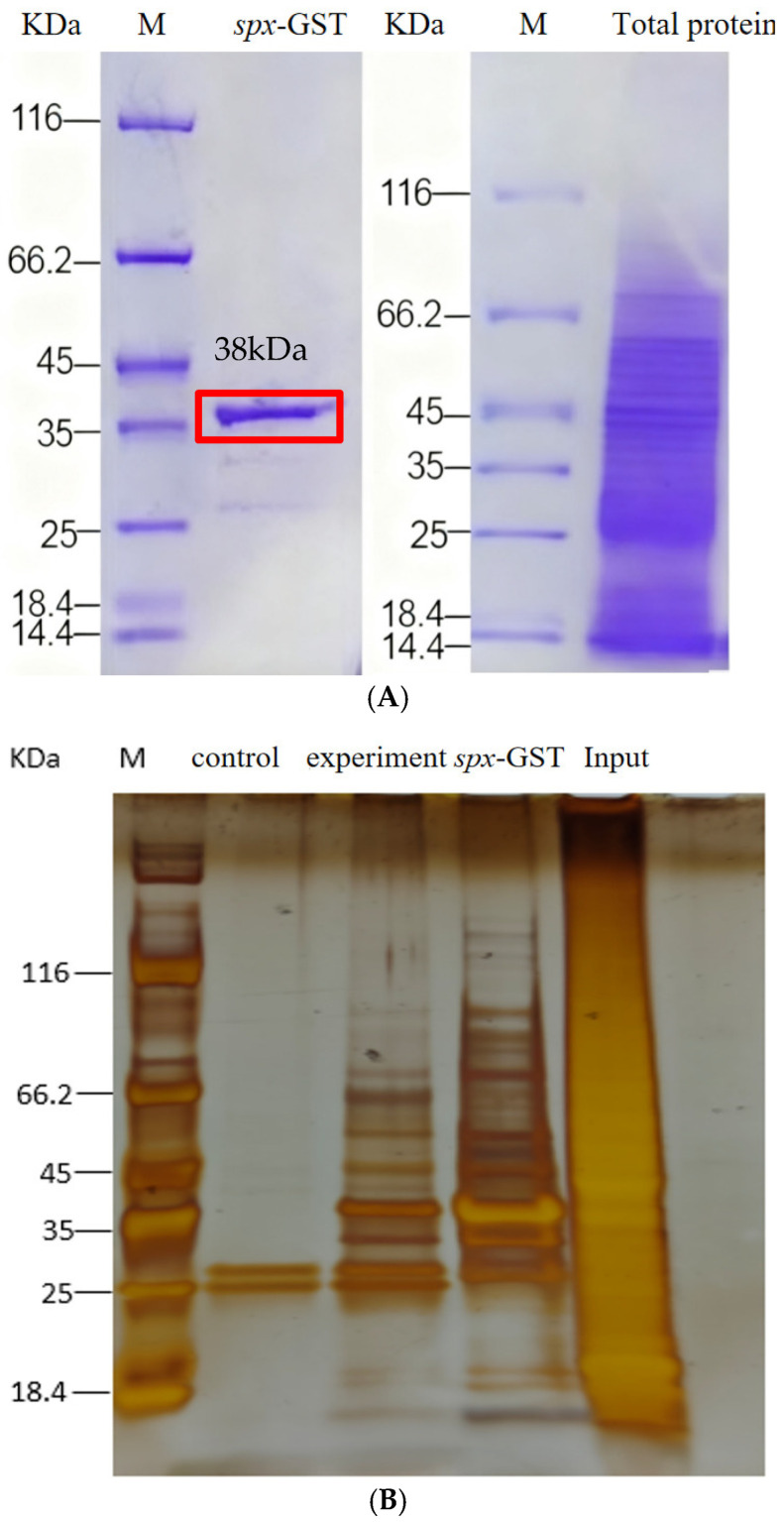
Isolation and identification of ScSPX-interacting proteins. (**A**) SDS-PAGE analysis of the expression of recombinant ScSPX in *E. coli*. (**B**) The fusion proteins GST or GST-ScSPX were used as bait, with the total protein extract from *S. chuatsi* serving as the input. The results are shown from a silver-stained SDS-PAGE gel.

**Figure 6 animals-15-02944-f006:**
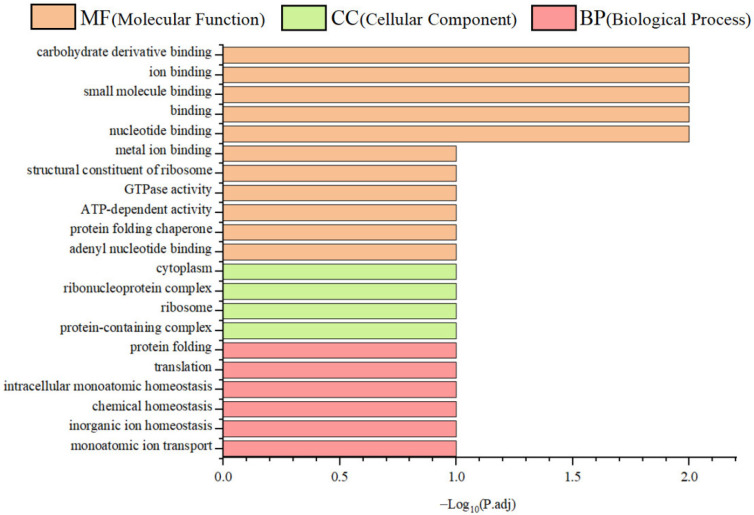
Gene ontology (GO) enrichment analysis of SPX-interacting proteins.

**Table 1 animals-15-02944-t001:** Primer sequence for *Siniperca chuatsi spx* gene cloning.

Primers	Sequences (5′–3′)	Purpose
*spx*-F1	CCCGTTTCATCACGTTGG	nested PCR
*spx*-R1	CATCACATTTCAAACATAGTCAGAG	nested PCR
*spx*-F2	ATGACGGGCTTGAGGAC	nested PCR
*spx*-R3	TCAGAAATATTCTCTCTTCCACACC	nested PCR
q*spx*-F	CGAAGGGCTCTTTCCAG	qPCR
q*spx*-R	CCTGCTGTAGGAAGTTGAG	qPCR
*β*-actin-F	GATCTGGCATCACACCTTCTAC	qPCR
*β*-actin-R	TCTTCTCCCTGTTGGCTTTG	qPCR
*β-Actin-F*	GCAAGCAGGAGTACGATGAGTC	qPCR
*β-Actin-R*	CAGTCGTTTGGGTTTGTAGCAG	qPCR

qPCR, quantitative polymerase chain reaction.

**Table 2 animals-15-02944-t002:** Information on candidate proteins that interact with SPX proteins.

Protein ID	Gene	Mass KDa	Score	Sequence Coverage %
G7Z090	Beta-actin (*ACTb*)	41.85	211.69	36.8
A0A0U1Z3U1	Heat shock protein 90beta (*HSP90beta*)	83.30	117.45	7.6
W8GMR6	Heat shock cognate 70-1 (*HSC70-1*)	71.28	47.50	9.5
W8GMB0	Heat shock cognate 70-2 (*HSC70-2*)	71.25	0.36	6.8
A0A7H1RFY2	RNA helicase (*DDX41*)	68.92	23.38	10.1
Q1A3S0	Myeloperoxidase (*MPO*)	85.73	70.49	10.4
A0A7H0S6W1	Cathepsin K (*ctsk*)	36.24	22.94	16.4
A0A0K2DRS2	Nonspecific cytotoxic cell receptor protein 1(*NCCRP-1*)	26.33	31.41	24.5
A0A889IUE0	Fatty acid-binding protein (*fabp2*)	15.15	24.96	29.5
A0A4Y5QVN6	60S ribosomal protein L13a (*rpl13a*)	26.27	8.73	11.4
F8UZB6	Cytochrome c oxidase subunit 2 (*COII*)	26.00	2.96	11.7
A0A289YPJ2	S-phase kinase-associated protein 1 (*Skp1*)	18.68	2.32	7.4
E2IFW3	Natural resistance-associated macrophage protein (*Nramp*)	61.44	0.98	1.4
MX	Interferon-induced GTP-binding protein Mx (*mx*)	71.43	0.53	1.6
A0A3G1HT24	60 kDa heat shock protein, mitochondrial (*HSP60*)	61.11	2.51	3.1
A0A5B8KAL3	Y-box binding protein-1 (*YBX1*)	34.27	1.47	3.5
A0A889IUT4	Fatty acid-binding protein (*fabp2a*)	15.22	1.37	6.8

## Data Availability

The raw data supporting the conclusions of this article will be made available by the authors, without undue reservation.

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
