# Peer review of "Spexin-Mediated Dietary Adaptation in Siniperca chuatsi: Molecular Characterisation and Functional Insights into FABP2 Interaction"

_animals, 2025, doi:10.3390/ani15202944_

Round 1

Reviewer 1 Report

Comments and Suggestions for Authors

General comments

The research topic has certain innovation and application value, especially in understanding the feeding adaptation mechanism of Siniperca chuatsi and promoting its artificial feed domestication. The experimental design is relatively complete and the amount of data is sufficient. However, there are some problems, such as the lack of in-depth key scientific problems, excessive speculation of data interpretation, and insufficient method details.

Specific comments

  1. The GST pull-down assay suggests an interaction between SPX and FABP2, but no further validation (e.g., Co-IP, SPR, or in vivo assays) is performed. There is no evidence that this interaction affects lipid metabolism or feeding behavior.
  2. The proposed “SPX–AMPK–PPAR–FABP2 axis” is highly speculative and lacks experimental support. The authors should clearly distinguish between hypothesis and evidence. It seems that there is failure to detect the AMPK activity, failure to verify the association between PPAR and FABP2, and failure to detect the lipid metabolism indicators.
  3. The difference in spx expression between large and small fish is interesting, but no physiological data (e.g., feed intake, fat content, metabolic rate) are provided to support the biological significance of this difference.
  4. Figure 5 (SDS-PAGE) is poorly labeled; lanes, protein sizes, and controls should be clearly indicated.
  5. The number of repetitions of the pull-down experiment is not specified (only "n=5 per time point" is mentioned, which refers to the number of animal samples rather than experimental replicates). Mass spectrometry results may be affected by accidental contamination, so at least 3 independent pull-down experiments are required to verify the stable interaction of FABP2;
  6. As a carnivorous fish, chuatsi may exhibit circadian rhythms in feeding behavior. However, the study does not specify the sampling time (e.g., whether sampling was conducted uniformly during the day or night). If spx expression is regulated by circadian rhythms, this will lead to deviations in the results;
  7. The study mentions that "each phenotype cohort included representative males and females", but the male-to-female ratio is not clearly stated (e.g., whether the large-body group had a 1:1 male-to-female ratio and whether the small-body group was matched). Additionally, the effect of gender on spx expression was not detected (it is known that sex-related hormones in fish may regulate the expression of metabolic genes), making it impossible to rule out the interference that "differences in spx expression are caused by gender rather than body size/feeding adaptation";
  8. Comparison of spx expression between large- and small-body groups should use independent samples t-test or Welch’s t-test (if variances are heterogeneous), but the study only mentions "P<0.05" without specifying the specific method, lacking detailed support.
  9. The discussion mentions that FABP2 is involved in lipid metabolism, but no literature is cited to explain the role of FABP2 in fish feeding or starvation response, leading to a lack of literature support for the conclusion that "the SPX-FABP2 interaction is related to feeding adaptation";
  10. The study mentions that "Scspx is highly expressed in the liver, consistent with Acipenser baerii but different from Scatophagus argus", but does not analyze the reasons for the differences (e.g., whether dietary differences between different fish species lead to different spx expression patterns), resulting in insufficient discussion depth.
  11. Missing Gene Function Verification, No Causal Evidence for Conclusions: The study only infers the function of Scspx based on expression correlations (e.g., spx expression changes during starvation, spx expression differences between body size groups), but no functional verification experiments have been conducted.

Author Response

For research article

Response to Reviewer 1 Comments

1. Summary

2. Questions for General Evaluation

Reviewer’s Evaluation

Response and Revisions

Does the introduction provide sufficient background and include all relevant references?

Yes

Thank you for your recognition.

Are all the cited references relevant to the research?

Yes

Thank you for your acknowledgment.

Is the research design appropriate?

Yes

Thank you.

Are the methods adequately described?

Can be improved

The corresponding information has been added based on the description of the materials provided in the text.

Are the results clearly presented?

Yes

Thank you for your positive recognition

Are the conclusions supported by the results?

Can be improved

We have supplemented  the results in accordance with the reviewers' comments to support related research and preliminary work.

3. Point-by-point response to Comments and Suggestions for Authors

Comments 1: The GST pull-down assay suggests an interaction between SPX and FABP2, but no further validation (e.g., Co-IP, SPR, or in vivo assays) is performed. There is no evidence that this interaction affects lipid metabolism or feeding behavior.

Response 1: We thank the reviewer for this insightful suggestion, which aligns well with our planned future work. As noted in the manuscript, this study represents a preliminary exploration of the spexin gene. Our initial findings suggest its potential role in growth phenotype. We agree that in vivo experiments are a crucial next step, and we will perform these studies as suggested to further validate the interactions between SPX and FABP2.

Evidence indicates that SPX enhances fatty acid metabolism by upregulating the expression of key genes including FAT/CD36, CPT1, ACADM, PPAR-α, and PGC1-α[1]. This link is evidenced by findings in Atlantic salmon, where ligand-bound fatty acid-binding proteins interact with PPARα, whose activation subsequently upregulates lipid-metabolizing genes[2].

1. Liu, Y., Sun, L., Zheng, L., Su, M., Liu, H., Wei, Y., Li, D., Wang, Y., Dai, C., Gong, Y., Zhao, C., & Li, Y. (2020). Spexin protects cardiomyocytes from hypoxia-induced metabolic and mitochondrial dysfunction. Naunyn-Schmiedeberg's archives of pharmacology, 393(1), 25–33. https://doi.org/10.1007/s00210-019-01708-0

2. Venold, F. F., Penn, M. H., Thorsen, J., Gu, J., Kortner, T. M., Krogdahl, A., & Bakke, A. M. (2013). Intestinal fatty acid binding protein (fabp2) in Atlantic salmon (Salmo salar): Localization and alteration of expression during development of diet induced enteritis. Comparative biochemistry and physiology. Part A, Molecular & integrative physiology, 164(1), 229–240. https://doi.org/10.1016/j.cbpa.2012.09.009

Comments 2: The proposed “SPX–AMPK–PPAR–FABP2 axis” is highly speculative and lacks experimental support. The authors should clearly distinguish between hypothesis and evidence. It seems that there is failure to detect the AMPK activity, failure to verify the association between PPAR and FABP2, and failure to detect the lipid metabolism indicators.

Response 2: We thank the reviewer for this critical and insightful comment. We completely agree that the "SPX-AMPK-PPAR-FABP2 axis" proposed in our initial manuscript was highly speculative and presented without sufficient clarification that it represents a hypothesis rather than a conclusively demonstrated pathway. We sincerely apologize for this lack of clarity. However, we fully acknowledge that we did not provide direct experimental evidence for AMPK activity, the PPAR-FABP2 link, or comprehensive lipid metabolism indicators, as correctly pointed out by the reviewer. This study constitutes a preliminary investigation into the function of the spexin gene, and the findings should be regarded as hypothetical insights into its potential biological effects. We fully acknowledge that further experimental validation is required to substantiate these initial conclusions. In accordance with the valuable suggestions provided by the reviewers, we will undertake systematic functional validation in subsequent research, with a focus on AMPK activity assays, verification of the regulatory relationship between PPAR and FABP2, and comprehensive analysis of lipid metabolism parameters.

  It has been established that adiponectin (AdipoQ) negatively regulates SPX expression through the PPAR-PI3K pathway [1]. In addition to its critical role in lipid and glucose homeostasis, evidence also suggests a direct function for AdipoQ in the regulation of feeding behavior [2].

1. Zheng, Y.; Bai, J.; He, M. Functional interaction of Spexin and Adiponectin forming a local feedbackin goldfish hepatocytes for feeding regulation. J Endocr Soc. 2021, 5(Suppl 1): A47.

2. Zheng, Y.; Ye, C.; He, M.; Ko, W, K, W.; Chan, Y, W.; Wong, A, O, L. Goldfish adiponectin: (I) molecular cloning, tissue distribution, recombinant protein expression, and novel function as a satiety factor in fish model. Front Endocrinol. 2023,1283298. https://doi/10.3389/fendo.2023.1283298

Comments 3:The difference in spx expression between large and small fish is interesting, but no physiological data (e.g., feed intake, fat content, metabolic rate) are provided to support the biological significance of this difference.

Response 3: We thank the reviewer for this insightful comment. We completely agree that linking the differential expression of spx to physiological traits is crucial for understanding its biological significance in the growth heterosis of S. chuatsi. This is indeed an important aspect that would strengthen our study. In response to this valuable suggestion, we have now significantly revised the Discussion section to address this point more thoroughly. Specifically, we have cited relevant literature that established the role of SPX as a key regulator of food intake and energy metabolism in other fish species. We hypothesize that the observed lower spx expression in large-bodied fish might be associated with a disinhibition of feeding behavior or a more efficient energy metabolism, potentially contributing to their accelerated growth. We clearly state that this remains a hypothesis derived from our correlation data, and that future studies specifically designed to measure these physiological parameters alongside spx expression are needed for validation. We have explicitly listed the investigation of the relationship between spx expression, feed intake, and metabolic rate as a key perspective for our future research.

Comments 4: Figure 5 (SDS-PAGE) is poorly labeled; lanes, protein sizes, and controls should be clearly indicated.

Response 4: Thank you for this critical observation and for giving us the opportunity to improve the clarity of our data presentation. We completely agree that the inadequate labeling in the original Figure 5 hindered the interpretation of the SDS-PAGE results. We sincerely apologize for this oversight. In direct response to your comment, we have thoroughly revised Figure 5 to include clear labels for all lanes, molecular weight markers, and controls. The specific modifications are as follows: Isolation and identification of Spx-interacting proteins. SDS-PAGE analysis of recombinant ScSPX protein expressed in E. coli. Following expression, a pull-down assay was conducted using GST-SPX as bait and total protein extracts from mandarin fish mixed tissues as input. The SDS-PAGE gel was visualized by silver nitrate staining. M:Marker.

The key bands of the pre-stained protein ladder are now annotated with their molecular weights (in kDa) on the gel image itself.

Comments 5: The number of repetitions of the pull-down experiment is not specified (only "n=5 per time point" is mentioned, which refers to the number of animal samples rather than experimental replicates). Mass spectrometry results may be affected by accidental contamination, so at least 3 independent pull-down experiments are required to verify the stable interaction of FABP2;

Response 5:We sincerely thank the reviewer for this valuable and constructive comment. We fully agree that the issue raised regarding the necessity of multiple independent pull-down experiments to verify the stable interaction of FABP2 is critical. We acknowledge that mass spectrometry results can indeed be affected by non-specific binding or accidental contamination, and performing at least three independent experiments is the gold standard for confirming robust protein-protein interactions.

To provide the most compelling evidence, we will perform three additional, entirely independent pull-down experiments. These experiments will be conducted starting from fresh cell cultures or animal tissue samples on separate days, using different reagent batches (including a new aliquot of antibody) for the immunoprecipitation, washing, elution, and mass spectrometry detection steps. We will specifically assess whether the interaction between FABP2 and the target protein is consistently reproduced across all three independent replicates.

We are confident that these supplemental experiments will effectively rule out the possibility of stochastic events and provide robust data support for the FABP2 interaction. Following your recommendation, we will proceed with the further validation experiments.

Comments 6: As a carnivorous fish, chuatsi may exhibit circadian rhythms in feeding behavior. However, the study does not specify the sampling time (e.g., whether sampling was conducted uniformly during the day or night). If spx expression is regulated by circadian rhythms, this will lead to deviations in the results;

Response 6: We sincerely thank the reviewer for raising this critical point. We completely agree that diurnal variations are a crucial factor to consider in physiological and molecular studies, especially for carnivorous fish like S. chuatsi with potentially strong feeding rhythms. In our study, we strictly controlled the sampling time to minimize this potential bias. All sampling procedures, for both the fasting groups and the control group, were conducted uniformly between 9:00 and 10:00 AM to control for potential diurnal fluctuations.

This information has now been explicitly added as followed: To minimize the potential influence of circadian rhythms on spexin (SPX) gene expression, the sampling time for all experimental groups (including the various fasting groups and the control group) was strictly restricted to between 9:00 and 10:00 AM each day. Therefore, we believe that the observed differences in spx expression are primarily attributable to the experimental treatments  rather than circadian effects.

Comments 7: The study mentions that "each phenotype cohort included representative males and females", but the male-to-female ratio is not clearly stated (e.g., whether the large-body group had a 1:1 male-to-female ratio and whether the small-body group was matched). Additionally, the effect of gender on spx expression was not detected (it is known that sex-related hormones in fish may regulate the expression of metabolic genes), making it impossible to rule out the interference that "differences in spx expression are caused by gender rather than body size/feeding adaptation"

Response 7: We sincerely thank the reviewer for this insightful and critical comment. The following description has been added to the Materials and Methods section of the manuscript:  Following dissection and sex identification, the subjects were randomly assigned to groups with a 1:1 male-to-female ratio, thereby minimizing any confounding effects attributable to sex.

Comments 8: Comparison of spx expression between large- and small-body groups should use independent samples t-test or Welch’s t-test (if variances are heterogeneous), but the study only mentions "P<0.05" without specifying the specific method, lacking detailed support.

Response 8:We thank the reviewer for this critical comment regarding the need for precise reporting of statistical methods. We completely agree that transparency in statistical analysis is fundamental for reproducibility, and we apologize for the oversight in not specifying the test used in our original manuscript. we have explicitly stated the specific statistical test used in the revised Statistics section of the Materials and Methods: “ An independent-sample t-test was used to compare the differences in spx expression between the large- and small-body size groups of S. chuatsi, differences were considered significant at P < 0.05.”

Comments 9: The discussion mentions that FABP2 is involved in lipid metabolism, but no literature is cited to explain the role of FABP2 in fish feeding or starvation response, leading to a lack of literature support for the conclusion that "the SPX-FABP2 interaction is related to feeding adaptation";

Response 9We thank the reviewer for the excellent suggestion to cite literature on the role of FABP2 in fish feeding and starvation responses, which will strengthen the support for our hypothesis on the SPX-FABP2 interaction.

Accordingly, we have revised the Discussion section by adding the following supporting references: Kaitetzidou et al. established a fasting-refeeding experimental model in gilthead sea bream and European sea bass, systematically revealing a dynamic regulatory relationship between FABP2 gene expression and nutritional status in fish [1]. The study demonstrated that fasting significantly down-regulated FABP2 mRNA expression in the intestinal tissues of both species, while refeeding promptly restored expression to baseline levels. This dynamic expression pattern indicates that FABP2 is directly regulated by the nutritional state of the organism. Notably, this regulatory mechanism appears highly conserved across teleost species. Similarly, Xia et al. observed in Asian sea bass that short-term fasting (3d, 6d, 12d) markedly suppressed the expression of both fabp2a and fabp2b genes in the intestine [2]. Furthermore, studies in zebrafish confirmed that increasing dietary lipid content significantly up-regulated intestinal fabp2 transcript levels [3]. These cross-species findings collectively suggest that FABP2 plays a critical role in nutrient metabolism in teleosts, likely participating in the regulation of lipid metabolic pathways through similar molecular mechanisms across different fish species, thereby supporting the view that this protein serves a core function in lipid absorption and transport.

1. Kaitetzidou, E.; Chatzifotis, S.; Antonopoulou, E.; Sarropoulou, E. Identification, Phylogeny, and Function of fabp2 Paralogs in Two Non-Model Teleost Fish Species. Marine biotechnology (New York, N.Y.). 2015, 17(5), 663–677. https://doi.org/10.1007/s10126-015-9648-6

2. Xia, J. H.; Lin, G.; He, X.; Liu, P.; Liu, F.; Sun, F.; Tu, R.; Yue, G. H. Whole genome scanning and association mapping identified a significant association between growth and a SNP in the IFABP-a gene of the Asian seabass. BMC genomics. 2013, 14, 295. https://doi.org/10.1186/1471-2164-14-295

3. Venkatachalam, A. B.; Sawler, D. L.; Wright, J. M. Tissue-specific transcriptional modulation of fatty acid-binding protein genes, fabp2, fabp3 and fabp6, by fatty acids and the peroxisome proliferator, clofibrate, in zebrafish (Danio rerio). Gene.2013, 520(1), 14–21. https://doi.org/10.1016/j.gene.2013.02.034

Comments 10: The study mentions that "Scspx is highly expressed in the liver, consistent with Acipenser baerii but different from Scatophagus argus", but does not analyze the reasons for the differences (e.g., whether dietary differences between different fish species lead to different spx expression patterns), resulting in insufficient discussion depth.

Response 10We are grateful to the reviewer for this exceptionally insightful comment. The reviewer rightly points out a missed opportunity to delve deeper into the potential evolutionary and physiological significance of the differential expression patterns of spx across fish species. We agree that exploring the reasons behind these differences, such as dietary habits, would greatly strengthen the discussion.

The following explanatory statements have been added to the text to address the underlying reasons: The observed variations in spx expression across species and tissues may be attributed to ecological adaptations associated with energy metabolism. As typical carnivorous fish, S. chuatsi and A. baerii exhibit metabolic traits adapted to high-protein and high-lipid diets, resulting in elevated energy metabolic demands on organs such as the liver and intestine. The high expression of spx in these tissues aligns with its putative roles in appetite suppression and lipid metabolism regulation, suggesting an evolutionary adaptation to a carnivorous ecological niche. In contrast, the omnivorous S. argus has a broader dietary spectrum, which may have driven the evolution of distinct energy utilization strategies. Such ecological divergence likely underlies the differential spx expression profiles: in S. argus, spx expression is diminished in conventional metabolic tissues but enhanced in regions associated with other physiological processes.

Comments 11: Missing Gene Function Verification, No Causal Evidence for Conclusions: The study only infers the function of Scspx based on expression correlations (e.g., spx expression changes during starvation, spx expression differences between body size groups), but no functional verification experiments have been conducted.

Response 11We sincerely thank the reviewer for this critical and insightful comment. We fully agree that functional validation experiments, are essential to establish a direct causal relationship for Scspx's role in metabolism. This is indeed a recognized limitation of the current study.

The primary aim of this research was to conduct an initial characterization of the Scspx gene in Siniperca chuatsi and to explore its correlation with metabolic states. While our data on expression patterns under starvation and in different size groups provide strong correlative evidence supporting the hypothesis that Scspx is involved in energy metabolism, we acknowledge that they do not constitute direct causal proof. Our findings echo the research direction of Zhao et al. (2025) regarding the potential function of Spexin (SPX1) in seahorse reproduction, thereby paving the way for future studies aimed at optimizing fish growth performance through SPX modulation.

To address this limitation directly and enhance the rigor of our manuscript, we have made the following modifications to the text: While the current study provides compelling evidence for the association between Scspx expression and growth characteristics in S. chuatsi, it is important to acknowledge its limitations. The functional inferences regarding Scspx are primarily derived from correlative observations in expression profiling. To advance from correlation to causation, direct functional validation is essential. Future investigations utilizing gene knockout/knockdown or overexpression approaches, both in vitro and in vivo, will be crucial to unequivocally establish the physiological role of Scspx in regulating lipid metabolism and its interaction with FABP2. Such studies will ultimately determine whether the observed expression changes represent a cause or a consequence of the metabolic alterations.

1. Zhao, L., Li, Y., Li, J., Jin, W., Chen, J., Wang, B. (2025). Molecular identification and reproductive function of spexin in the big-belly seahorse (Hippocampus abdominalis). General and comparative endocrinology, 367, 114721. https://doi.org/10.1016/j.ygcen.2025.114721

4. Response to Comments on the Quality of English Language

The English is fine and does not require any improvement.

5. Additional clarifications

Nope.

Reviewer 2 Report

Comments and Suggestions for Authors

this is might be a valuable manuscript on the tissue specific expression of spexin,spx, in a single fish species. such results are very needed nowadays.

i have identified 2 major issues:

1.in one of M&M paragraphs, authors say they used "brain" for their qRT-PCR, in another paragraph, they say they used "telencephalon" for the qRT-PCR. hence, what part of brain was used? why did not authors used diencephalon that is known to harbour the centres regulating food intake as they even cite in other species? it would not be easy to separate hypothalamus, which is the main centre of the food intake control, however, at least the diencephalon should have been used instead of telencephalon or the whole "brain" as probably done by the authors but not specified. the reasons are that 1.)telencephalon does NOT contain all the brain regions relevant for this study, and 2.) the whole-brain RNA is kind of a chaos...

2.the authors should exactly and clearly specify how many individuals were used for each qRT-PCR and other experiments in the manuscript main text as well as in each graph/image. in M&M they somewhere mentioned that 5 individuals were used but it is uncleary, how many individuals yielded sufficent amount and quality of RNA for each experiment/reaction. this is crucialy needed.

a minor issue is lack of references in the Introduction section lines 48-55 and the same is missing in the entire paragraph in Discussion l.346-356 - actually, both parts are on similar topics.

Comments on the Quality of English Language

several formal issues detected, can be improved easily

Author Response

For research article

Response to Reviewer 2 Comments

1. Summary

We appreciate the time and effort you have dedicated to reviewing our manuscript. Your insightful comments have been highly valuable. We have addressed all the points raised in the responses below, and the corresponding revisions have been incorporated into the manuscript, which is highlighted using the track-changes function.

2. Questions for General Evaluation

Reviewer’s Evaluation

Response and Revisions

Does the introduction provide sufficient background and include all relevant references?

Can be improved

We have provided sufficient background information and included all relevant references in the introduction section based on your feedback.

Are all the cited references relevant to the research?

Yes

Thank you for your recognition.

Is the research design appropriate?

Can be improved

We have strengthened the research design accordingly, as detailed in the revised Methodology section

Are the methods adequately described?

Must be improved

We have thoroughly revised the Methods section to provide a more detailed and reproducible description of the experimental procedures, as requested

Are the results clearly presented?

Can be improved

We have revised the Results section

Are the conclusions supported by the results?

Can be improved

We have carefully revised the Conclusions section

3. Point-by-point response to Comments and Suggestions for Authors

Comments 1: in one of M&M paragraphs, authors say they used "brain" for their qRT-PCR, in another paragraph, they say they used "telencephalon" for the qRT-PCR. hence, what part of brain was used? why did not authors used diencephalon that is known to harbour the centres regulating food intake as they even cite in other species? it would not be easy to separate hypothalamus, which is the main centre of the food intake control, however, at least the diencephalon should have been used instead of telencephalon or the whole "brain" as probably done by the authors but not specified. the reasons are that 1.)telencephalon does NOT contain all the brain regions relevant for this study, and 2.) the whole-brain RNA is kind of a chaos.

Response 1: We thank the reviewer for this important question. The use of different brain regions was intentional and based on the distinct objectives of each experiment. For the initial tissue distribution analysis, we used the whole brain as this approach is widely adopted in the literature to provide a broad overview of neuropeptide expression patterns (Zhang et al. , 2024; Tian et al. , 2024)

For the functional study investigating the role of SPX in hunger and dietary adaptation, we specifically micro-dissected the telencephalon. This region was targeted based on prior studies implicating it in the regulation of feeding behavior and metabolic adaptation in teleost fish (Wong et al., 2013).

We agree that analyzing specific hypothalamic nuclei would be highly informative, and this represents an important direction for our future research.

Zhang, Y., Wang, J., Yang, L., Yan, X., Qin, C., & Nie, G. (2024). Spexin acts as a novel glucose-lowering factor in grass carp (Ctenopharyngodon idella). Biochemical and biophysical research communications, 708, 149810. https://doi.org/10.1016/j.bbrc.2024.149810

Tian, Z., Yu, Z., Xu, Y., Cui, A., Jiang, Y., Huang, H., & Wang, B. (2024). Spexin and its receptors in the yellowtail kingfish (Seriola lalandi): identification, expression profiles and reproductive function. Fish physiology and biochemistry, 50(6), 2453–2474. https://doi.org/10.1007/s10695-024-01394-7

Wong, M. K., Sze, K. H., Chen, T., Cho, C. K., Law, H. C., Chu, I. K., & Wong, A. O. (2013). Goldfish spexin: solution structure and novel function as a satiety factor in feeding control. American journal of physiology. Endocrinology and metabolism, 305(3), E348–E366. https://doi.org/10.1152/ajpendo.00141.2013

Comments 2: the authors should exactly and clearly specify how many individuals were used for each qRT-PCR and other experiments in the manuscript main text as well as in each graph/image. in M&M they somewhere mentioned that 5 individuals were used but it is uncleary, how many individuals yielded sufficent amount and quality of RNA for each experiment/reaction. this is crucialy needed.

Response 2: We appreciate the reviewer for highlighting this important point. The description in the Materials and Methods section has been clarified to specify the number of biological replicates (n) used in each experiment. The supplementary information is provided below: Experimental groups underwent starvation conditioning with sampling at 3, 5, and 7 days post-starvation, with five fish randomly collected from the net cage at each time point. A total of five mandarin fish (S. chuatsi) maintained under routine feeding conditions were selected from the pond.

a minor issue is lack of references in the Introduction section lines 48-55 and the same is missing in the entire paragraph in Discussion l.346-356 - actually, both parts are on similar topics.

Response : We sincerely appreciate your valuable suggestion. We agree that the lack of citations in the Introduction (lines 48-55) and the Discussion (lines 346-356) weakened the strength of our arguments. Accordingly, we have thoroughly revised these sections by adding key references as detailed below.

Introduction (Lines 48-55):

1. Wu, H.; Lin, F.; Chen, H.; Liu, J.; Gao, Y.; Zhang, X.; Hao, J.; Chen, D.; Yuan, D.; Wang, T.; Li, Z. Ya-fish (Schizothorax prenanti) spexin: identification, tissue distribution and mRNA expression responses to periprandial and fasting. Fish Physiol Biochem. 2016, 42(1), 39–49. https://doi/10.1007/s10695-015-0115-0

2. Wang, S.; Wang, B.; Chen, S. Spexin in the half-smooth tongue sole (Cynoglossus semilaevis): molecular cloning, expression profiles, and physiological effects. Fish physiology and biochemistry. 2018, 44(3), 829–839. https://doi.org/10.1007/s10695-018-0472-6

3. Deng, S. P.; Chen, H. P.; Zhai, Y.; Jia, L. Y.; Liu, J. Y.; Wang, M.; Jiang, D. N.; Wu, T. L.; Zhu, C. H.; Li, G. L. Molecular cloning, characterization and expression analysis of spexin in spotted scat (Scatophagus argus). Gen Comp Endocrinol. 2018, 266, 60–66. https://doi/10.1016/j.ygcen.2018.04.018

Discussion (Lines 346-356):

1. Wu, H.; Lin, F.; Chen, H.; Liu, J.; Gao, Y.; Zhang, X.; Hao, J.; Chen, D.; Yuan, D.; Wang, T.; Li, Z. Ya-fish (Schizothorax prenanti) spexin: identification, tissue distribution and mRNA expression responses to periprandial and fasting. Fish Physiol Biochem. 2016, 42(1), 39–49. https://doi/10.1007/s10695-015-0115-0

2. Tian, Z.; Xu, S.; Wang, M.; Li, Y.; Chen, H.; Tang, N.; Wang, B.; Zhang, X.; Li, Z. Identification, tissue distribution, periprandial expression, and anorexigenic effect of spexin in Siberian sturgeon, Acipenser baeri. Fish Physiol Biochem. 2020, 46(6), 2073–2084. https://doi/10.1007/s10695-020-00856-y

3. Wong, M. K.; Sze, K. H.; Chen, T.; Cho, C. K.; Law, H. C.; Chu, I. K.; Wong, A. O. Goldfish spexin: solution structure and novel function as a satiety factor in feeding control. Am J Physiol Endocrinol Metab. 2013, 305(3), E348–E366. https://doi/10.1152/ajpendo.00141.2013

4. Response to Comments on the Quality of English Language

Point 1: The English could be improved to more clearly express the research

Response 1: The language of our manuscript has been polished by a professional editing service.

5. Additional clarifications

Nope.

For review article

Response to Reviewer X Comments

1. Summary

Thank you very much for taking the time to review this manuscript. Please find the detailed responses below and the corresponding revisions/corrections highlighted/in track changes in the re-submitted files. [This is only a recommended summary. Please feel free to adjust it. We do suggest maintaining a neutral tone and thanking the reviewers for their contribution although the comments may be negative or off-target. If you disagree with the reviewer's comments please include any concerns you may have in the letter to the Academic Editor.]

2. Questions for General Evaluation

Reviewer’s Evaluation

Response and Revisions

Is the work a significant contribution to the field?

[Please give your response if necessary. Or you can also give your corresponding response in the point-by-point response letter. The same as below]

Is the work well organized and comprehensively described?

Is the work scientifically sound and not misleading?

Are there appropriate and adequate references to related and previous work?

Is the English used correct and readable?

3. Point-by-point response to Comments and Suggestions for Authors

Comments 1: [Paste the full reviewer comment here.]

Response 1: [Type your response here and mark your revisions in red] Thank you for pointing this out. I/We agree with this comment. Therefore, I/we have.[Explain what change you have made. Mention exactly where in the revised manuscript this change can be found – page number, paragraph, and line.]

“[updated text in the manuscript if necessary]”

Comments 2: [Paste the full reviewer comment here.]

Response 2: Agree. I/We have, accordingly, done/revised/changed/modified…..to emphasize this point. Discuss the changes made, providing the necessary explanation/clarification. Mention exactly where in the revised manuscript this change can be found – page number, paragraph, and line.]

“[updated text in the manuscript if necessary]”

4. Response to Comments on the Quality of English Language

Point 1:

Response 1:    (in red)

5. Additional clarifications

Nope.

Reviewer 3 Report

Comments and Suggestions for Authors

The aim of this study is to identify the Spexin (spx) gene in Siniperca chuatsi and investigate its role in mediating dietary adaptation (starvation and refeeding), particularly through its interaction with fatty acid binding protein 2 (FABP2). The topic of the article has a certain degree of innovation. For the first time, neuropeptide spx is linked with fatty acid transporter FABP2 in the regulation of fish nutrition metabolism. The technical route design is relatively comprehensive, combining molecular cloning, gene expression analysis, protein interaction verification, and transcriptional regulation analysis. However, despite the novelty of the research direction, the manuscript has significant shortcomings in terms of the rigor of experimental design, the completeness of the evidence chain for some key conclusions, and the depth of interpretation of the results. Firstly, the core mechanism of the study is not thoroughly demonstrated. Although the physical interaction between spx and FABP2 has been confirmed, the functional consequences of this interaction have not been elucidated. Secondly, there is a critical disconnect in the research design, where physiological observations in vivo (starvation/refeeding) and molecular interactions/transcriptional regulation analysis in vitro have not been effectively integrated to form a coherent biological story. In addition, some data analysis and result interpretation are superficial, and the discussion section fails to fully explore the inherent connections of the data. Therefore, the current status of this article does not meet the publication requirements and requires significant revisions.

  1. Abstract

Problem: Although the abstract summarizes the main experimental content, it fails to highlight the core innovation of the research. The result descriptions are mostly qualitative statements, lacking key quantitative data support. The uniqueness of this study should be more clearly stated in the abstract, which reveals for the first time the direct interaction between spx and FABP2 in fish and their potential co regulatory effects on downstream genes.

  1. Introduction

Problem: The background introduction is relatively sufficient, but the analysis of the limitations of existing research is not in-depth enough, resulting in the entry point of this study not being prominent enough.

Suggestion:

The introduction should provide a more in-depth analysis of the shortcomings in current research on spx in regulating fish metabolism. For example, have existing studies mainly focused on its regulation of feeding, while neglecting its direct role in the metabolism of nutrients (such as fatty acids) at the cellular level? This can better highlight the innovation of this study in connecting spx and FABP2.

At the end of the introduction, the core scientific hypothesis that this study aims to verify should be presented more clearly.

  1. Materials and Methods

Problem: There are obvious deficiencies in this section. Incomplete ethical approval information; The key experimental design, especially the lack of methods to demonstrate the functional consequences of protein interactions; The description of statistical methods is too general.

Suggestion:

Ethical approval: The article mentions that animal experiments have been approved, but a specific ethics committee approval number must be provided.

Experimental design flaws:

Lack of functional consequence verification: The core of the study is the interaction between spx and FABP2, but the author only demonstrated that the two can be combined without exploring the functional significance of this combination. For example, does the binding of spx affect the activity of FABP2 in transporting fatty acids? Or, does FABP2 act as a "chaperone protein" to transport spx to specific locations for its function? Additional experiments must be conducted to answer these key questions, otherwise the depth of the research will be greatly reduced.

In vitro and in vivo experiments were disconnected: The co expression pattern of spx and FABP2 was observed in the hunger/refeeding experiment, but subsequent molecular experiments (Co IP, pull-down, dual luciferase) were conducted in the overexpression system (HEK293T cells). Additional experiments should be conducted to demonstrate that this interaction and transcriptional regulation also exist endogenously in fish cells.

Statistical method: The statistical software used, version, and whether the normal distribution and homogeneity of variance of the data were tested before conducting t-tests or ANOVA analysis should be explained in detail.

  1. Results

Problem: The quality of the charts is average, and the logical correlation of some results is not strong. The analysis of protein interactions and transcriptional regulation is relatively preliminary.

Suggestion:

Improve chart quality: The font size and resolution of all charts should be optimized. The images of Western Blot and Co IP should display the complete Marker lane and provide higher quality images with cleaner backgrounds.

Insufficient evidence: The dual luciferase reporter assay in Figure 5 showed that co transfection of spx and FABP2 had an impact on the promoter activity of some target genes, such as adipoq. However, this does not directly prove that spx or FABP2 are transcription factors. A more likely scenario is that their interaction affects the activity of a genuine transcription factor. The current conclusion has a jump.

Logical disconnect: The results section jumps from gene cloning and expression analysis to protein interactions, and then to transcriptional regulation analysis of seemingly unrelated target genes (adipoq, leptin). There is a lack of a clear logical thread to connect these parts. The author needs to explain why these specific downstream genes were selected for testing

  1. Discussion

Problem: There is a lot of repetitive description of the results in the discussion, and the exploration of the mechanism is very limited. The experimental results from different parts have not been integrated into a convincing biological model.

Suggestion:

The restatement of the results should be significantly reduced. The focus of the discussion should be to propose a clear * * Working Model * * to explain how the expression of spx and FABP2 changes under hunger or refeeding stress, how they interact, and how this interaction ultimately affects the fat metabolism and energy homeostasis of fish?

In depth exploration of mechanisms: The discussion should focus on the functional significance of the interaction between spx-FABP2. The author should propose several possible hypotheses based on existing data and literature, such as the functional or positional effects mentioned earlier, and point out how these hypotheses need to be validated through experiments in the future.

Objective Explanation of Limitations: The author should candidly analyze the limitations of this study in the discussion, particularly the lack of direct verification of the functional consequences of interactions and the disconnect between in vivo and in vitro experiments.

  1. Language&Format

Problem: There are some grammar errors and awkward expressions in English, which affect the fluency of reading. The reference format needs to be carefully checked.

Suggestion:

It is recommended that the manuscript undergo thorough language polishing by a professional native English editor before resubmitting.

Strictly follow the format requirements of the journal, unify the layout, chart citation, and reference citation format of the entire text.

Author Response

Response to Reviewer 3 Comments

1. Summary

2. Questions for General Evaluation

Reviewer’s Evaluation

Response and Revisions

Does the introduction provide sufficient background and include all relevant references?

Can be improved

We have provided sufficient background information and included all relevant references in the introduction section based on your feedback.

Are all the cited references relevant to the research?

Yes

Thank you for your acknowledgment.

Is the research design appropriate?

Yes

Thank you.

Are the methods adequately described?

Yes

Thank you for your recognition.

Are the results clearly presented?

Yes

Thank you for your positive recognition

Are the conclusions supported by the results?

Yes

Thank you for your recognition.

3. Point-by-point response to Comments and Suggestions for Authors

Comments 1: Problem: Although the abstract summarizes the main experimental content, it fails to highlight the core innovation of the research. The result descriptions are mostly qualitative statements, lacking key quantitative data support. The uniqueness of this study should be more clearly stated in the abstract, which reveals for the first time the direct interaction between spx and FABP2 in fish and their potential co regulatory effects on downstream genes

Response 1: We sincerely thank the reviewer for this insightful and constructive comment. In response, we have thoroughly revised the abstract to enhance its clarity and impact, with the key improvements as follows: Our results provide the first evidence of a direct SPX-FABP2 interaction in fish, pointing to a coordinated role in downstream gene regulation. This work hereby uncovers a novel regulatory axis within the piscine energy metabolism network. 

To incorporate quantitative data, we have replaced qualitative statements with specific results. For instance: The predominant expression of spx was found in the liver of feed-trained S. chuatsi, where it was 17.36-fold greater than in muscle.

Furthermore, the abstract now emphasizes the size-dependent expression differences: "Compared to the smallest individuals, hepatic, and brain spx expression was substantially lower in the largest individuals, whereas stomach expression was higher(P<0.05)."

Comments 2: Problem: The background introduction is relatively sufficient, but the analysis of the limitations of existing research is not in-depth enough, resulting in the entry point of this study not being prominent enough.

Suggestion:

The introduction should provide a more in-depth analysis of the shortcomings in current research on spx in regulating fish metabolism. For example, have existing studies mainly focused on its regulation of feeding, while neglecting its direct role in the metabolism of nutrients (such as fatty acids) at the cellular level? This can better highlight the innovation of this study in connecting spx and FABP2.

At the end of the introduction, the core scientific hypothesis that this study aims to verify should be presented more clearly.

Response 2: We are grateful to the reviewer for this insightful suggestion. In response, we have thoroughly revised the introduction section as follows: However, the structure and function of the spx gene in S. chuatsi remain uncharacterized. Furthermore, although the roles of spexin in reproduction and feeding are well-established, its direct influence on growth variation and lipid metabolism in teleosts is poorly understood, particularly on a mechanistic basis. To investigate the potential role of the spx gene (Scspx) in nutrient metabolism regulation in mandarin fish (Siniperca chuatsi), we cloned Scspx from feed-adapted fish, profiled its tissue expression patterns, and examined its expression in response to starvation and in populations with divergent feed-adaptation phenotypes. Furthermore, we screened for potential interacting proteins to elucidate the regulatory function of Scspx in feeding and metabolic pathways during the feed adaptation process. 

Comments 3: Problem: There are obvious deficiencies in this section. Incomplete ethical approval information; The key experimental design, especially the lack of methods to demonstrate the functional consequences of protein interactions; The description of statistical methods is too general.

Lack of functional consequence verification: The core of the study is the interaction between spx and FABP2, but the author only demonstrated that the two can be combined without exploring the functional significance of this combination. For example, does the binding of spx affect the activity of FABP2 in transporting fatty acids? Or, does FABP2 act as a "chaperone protein" to transport spx to specific locations for its function? Additional experiments must be conducted to answer these key questions, otherwise the depth of the research will be greatly reduced.

In vitro and in vivo experiments were disconnected: The co-expression pattern of spx and FABP2 was observed in the hunger/refeeding experiment, but subsequent molecular experiments (Co IP, pull-down, dual luciferase) were conducted in the overexpression system (HEK293T cells). Additional experiments should be conducted to demonstrate that this interaction and transcriptional regulation also exist endogenously in fish cells.

Statistical method: The statistical software used, version, and whether the normal distribution and homogeneity of variance of the data were tested before conducting t-tests or ANOVA analysis should be explained in detail.

Response 3: We sincerely apologize for this omission. The 'Materials and Methods' section has been updated to include an explicit statement regarding ethical approval at its outset: "All experiments were carried out in accordance with the “Guidelines for the Protection and Use of Laboratory Animals in China.”All experimental procedures and sample collection were approved by the Animal Experimentation Ethics Committee of the Pearl River Fisheries Research, Chinese Academy of Fishery Sciences. "

As noted in the manuscript, this study represents a preliminary exploration of the spexin gene. Our initial findings suggest its potential role in growth phenotype. We agree that in vivo experiments are a crucial next step, and we will perform these studies as suggested to further validate the interactions between SPX and FABP2.

Evidence indicates that SPX enhances fatty acid metabolism by upregulating the expression of key genes including FAT/CD36, CPT1, ACADM, PPAR-α, and PGC1-α[1]. This link is evidenced by findings in Atlantic salmon, where ligand-bound fatty acid-binding proteins (fabp2) interact with PPARα, whose activation subsequently upregulates lipid-metabolizing genes[2]. 

1. Liu, Y., Sun, L., Zheng, L., Su, M., Liu, H., Wei, Y., Li, D., Wang, Y., Dai, C., Gong, Y., Zhao, C., & Li, Y. (2020). Spexin protects cardiomyocytes from hypoxia-induced metabolic and mitochondrial dysfunction. Naunyn-Schmiedeberg's archives of pharmacology, 393(1), 25–33. https://doi.org/10.1007/s00210-019-01708-0

2. Venold, F. F., Penn, M. H., Thorsen, J., Gu, J., Kortner, T. M., Krogdahl, A., & Bakke, A. M. (2013). Intestinal fatty acid binding protein (fabp2) in Atlantic salmon (Salmo salar): Localization and alteration of expression during development of diet induced enteritis. Comparative biochemistry and physiology. Part A, Molecular & integrative physiology, 164(1), 229–240. https://doi.org/10.1016/j.cbpa.2012.09.009

  We acknowledge the reviewer's valid comment regarding the initial lack of detail in our statistical reporting. In response, we have now expanded the 'Materials and Methods' section to provide a complete account of the statistical analyses performed, which is critical for interpreting the results with confidence.

Comments 4: Problem: The quality of the charts is average, and the logical correlation of some results is not strong. The analysis of protein interactions and transcriptional regulation is relatively preliminary.

Suggestion:

Improve chart quality: The font size and resolution of all charts should be optimized. The images of Western Blot and Co IP should display the complete Marker lane and provide higher quality images with cleaner backgrounds.

Insufficient evidence: The dual luciferase reporter assay in Figure 5 showed that co transfection of spx and FABP2 had an impact on the promoter activity of some target genes, such as adipoq. However, this does not directly prove that spx or FABP2 are transcription factors. A more likely scenario is that their interaction affects the activity of a genuine transcription factor. The current conclusion has a jump.

Logical disconnect: The results section jumps from gene cloning and expression analysis to protein interactions, and then to transcriptional regulation analysis of seemingly unrelated target genes (adipoq, leptin). There is a lack of a clear logical thread to connect these parts. The author needs to explain why these specific downstream genes were selected for testing

Response 4: We thank the reviewer for their valuable feedback on the quality of the figures. We fully agree that high-quality images are crucial for clearly communicating scientific data. The Western Blot and Co-IP experiments have not been performed in the current phase of this study and will be included in subsequent investigations. Additionally, in accordance with the reviewers' suggestions, we have optimized the background of the GST pull-down assay images and provided higher-quality figures.

  We are grateful to the reviewer for their meticulous review and valuable insights on Figure 5. We wish to clarify that this figure illustrates a GST pull-down assay followed by SDS-PAGE analysis, the objective of which was to confirm a direct protein-protein interaction between SPX and FABP2 under in vitro conditions. It was not intended to evaluate the regulation of gene promoter activity. We have meticulously revised all pertinent descriptions in the manuscript to eliminate this confusion. The fundamental conclusion derived from this finding is that the SPX-FABP2 interaction potentially plays a regulatory role in the expression of genes associated with lipid metabolism downstream.

   The research was conducted following the core logical pathway outlined below, which also systematically explains our research objectives and framework: Initially, gene cloning and expression analysis revealed that under hunger/refeeding stress conditions, the expression of the spx gene exhibited significant dynamic changes. This phenomenon suggested that this gene might play an important role in the processes of dietary adaptation and metabolic adaptability regulation in fish. To further elucidate the molecular mechanism by which the spx gene participates in dietary adaptation, we subsequently conducted a protein-protein interaction study. Results from the GST pull-down assay confirmed a direct physical interaction between SPX and FABP2 in vitro, providing a potential molecular basis for their functional synergy. This study provides experimental evidence clarifying the intrinsic relationship between spx expression and dietary adaptation, while we also acknowledge the limitations of the current research. The specific mechanism by which the formed SPX-FABP2 complex executes its biological function will be a key focus of our future research direction.

Comments 5: Problem: There is a lot of repetitive description of the results in the discussion, and the exploration of the mechanism is very limited. The experimental results from different parts have not been integrated into a convincing biological model.

Suggestion:

The restatement of the results should be significantly reduced. The focus of the discussion should be to propose a clear * * Working Model * * to explain how the expression of spx and FABP2 changes under hunger or refeeding stress, how they interact, and how this interaction ultimately affects the fat metabolism and energy homeostasis of fish?

In depth exploration of mechanisms: The discussion should focus on the functional significance of the interaction between spx-FABP2. The author should propose several possible hypotheses based on existing data and literature, such as the functional or positional effects mentioned earlier, and point out how these hypotheses need to be validated through experiments in the future.

Objective Explanation of Limitations: The author should candidly analyze the limitations of this study in the discussion, particularly the lack of direct verification of the functional consequences of interactions and the disconnect between in vivo and in vitro experiments.

Response 5: We are grateful to the reviewer for this constructive feedback. In response to the valid points raised regarding the repetitiveness of the Results section and the need for a conceptual framework, we have comprehensively restructured the Discussion. The revisions primarily involve: (1)substantially condensing the restatement of results, and  introducing a well-defined working model that synthesizes our key discoveries into a coherent narrative. We have revised the section as followed:

Kaitetzidou et al. established a fasting-refeeding experimental model in gilthead sea bream and European sea bass, systematically revealing a dynamic regulatory relationship between FABP2 gene expression and nutritional status in fish [1]. The study demonstrated that fasting significantly down-regulated FABP2 mRNA expression in the intestinal tissues of both species, while refeeding promptly restored expression to baseline levels. This dynamic expression pattern indicates that FABP2 is directly regulated by the nutritional state of the organism. Notably, this regulatory mechanism appears highly conserved across teleost species. Similarly, Xia et al. observed in Asian sea bass that short-term fasting (3d, 6d, 12d) markedly suppressed the expression of both fabp2a and fabp2b genes in the intestine [2]. Furthermore, studies in zebrafish confirmed that increasing dietary lipid content significantly up-regulated intestinal fabp2 transcript levels.The results suggest that the regulation of fabp2 gene transcription by fatty acids is mediated by the interaction of the peroxisome proliferator-activated receptor (PPAR) with a peroxisome proliferator response element (PPRE) in its promoter region [3]. Based on the observed upregulation of spx expression in the intestine of S. chuatsi subjected to short-term starvation after feeding, this study proposes a regulatory hypothesis wherein spx may indirectly facilitate fabp2 gene transcription via the upregulation of PPAR-α expression. This signaling cascade potentially represents a key mechanistic axis through which spx exerts its pivotal role in modulating lipid metabolism in fish.

1. Kaitetzidou, E.; Chatzifotis, S.; Antonopoulou, E.; Sarropoulou, E. Identification, Phylogeny, and Function of fabp2 Paralogs in Two Non-Model Teleost Fish Species. Marine biotechnology (New York, N.Y.). 2015, 17(5), 663–677. https://doi.org/10.1007/s10126-015-9648-6

2. Xia, J. H.; Lin, G.; He, X.; Liu, P.; Liu, F.; Sun, F.; Tu, R.; Yue, G. H. Whole genome scanning and association mapping identified a significant association between growth and a SNP in the IFABP-a gene of the Asian seabass. BMC genomics. 2013, 14, 295. https://doi.org/10.1186/1471-2164-14-295

3. Venkatachalam, A. B.; Sawler, D. L.; Wright, J. M. Tissue-specific transcriptional modulation of fatty acid-binding protein genes, fabp2, fabp3 and fabp6, by fatty acids and the peroxisome proliferator, clofibrate, in zebrafish (Danio rerio). Gene.2013, 520(1), 14–21. https://doi.org/10.1016/j.gene.2013.02.034

(2)To the best of our knowledge, no prior research has directly confirmed an interaction between Spx and Fabp proteins. Our study now establishes the first experimental evidence for a direct physical interaction between Spx and Fabp2 in fish. Based on this discovery, we outline the following future research directions: (1) At the cellular level, SPX knockdown or overexpression will be performed to assess its regulatory effect on Fabp2; (2) Subcellular localization will be examined via confocal microscopy to determine whether SPX co-expression alters the intracellular distribution of FABP2; (3) We aim to identify upstream regulatory signals of SPX, such as phosphorylation, and systematically investigate how these post-translational modifications dynamically modulate the SPX–FABP2 interaction under fasting and refeeding conditions.

(3)We concur that a candid exploration of these aspects not only strengthens the manuscript's rigor but also provides valuable direction for future investigations. In response, we have made the following specific changes: While the current study provides compelling evidence for the association between Scspx expression and growth characteristics in S. chuatsi, it is important to acknowledge its limitations. The functional inferences regarding Scspx are primarily derived from correlative observations in expression profiling. To advance from correlation to causation, direct functional validation is essential. Future investigations utilizing gene knockout/knockdown or overexpression approaches, both in vitro and in vivo, will be crucial to unequivocally establish the physiological role of Scspx in regulating lipid metabolism and its interaction with FABP2. Such studies will ultimately determine whether the observed expression changes represent a cause or a consequence of the metabolic alterations.

Comments 6: Problem: There are some grammar errors and awkward expressions in English, which affect the fluency of reading. The reference format needs to be carefully checked.

Suggestion:

It is recommended that the manuscript undergo thorough language polishing by a professional native English editor before resubmitting.

Strictly follow the format requirements of the journal, unify the layout, chart citation, and reference citation format of the entire text.

Response 6:We wish to express our sincere agreement with your comments on the grammatical errors, awkward expressions, and reference formatting inconsistencies. Based on your valuable recommendations, we have diligently performed a thorough revision of the manuscript.

4. Response to Comments on the Quality of English Language

Point 1: The English could be improved to more clearly express the research.

Response 1: The language of our manuscript has been polished by a professional editing service.   

5. Additional clarifications

Nope.

Round 2

Reviewer 1 Report

Comments and Suggestions for Authors

All my concerns have been addressed. No other comments.

Reviewer 2 Report

Comments and Suggestions for Authors

the authors have addressed my comments and i do not have any other

Reviewer 3 Report

Comments and Suggestions for Authors

The proposed review comments have been revised